# The Anti-Fibrotic Effects of CG-745, an HDAC Inhibitor, in Bleomycin and PHMG-Induced Mouse Models

**DOI:** 10.3390/molecules24152792

**Published:** 2019-07-31

**Authors:** Young-Suk Kim, Hyunju Cha, Hyo-Jin Kim, Joong-Myung Cho, Hak-Ryul Kim

**Affiliations:** 1Department of Internal Medicine, Institute of Wonkwang Medical Science, Wonkwang University, School of Medicine, Iksan, Jeonbuk 54538, Korea; 2Medical Convergence Research Center, Wonkwang University Hospital, Iksan, Jeonbuk 54538, Korea; 3CrystalGenomics, Inc. Bundanggu, Seongnamsi, Gyeonggido 13488, Korea

**Keywords:** CG-745, HDAC inhibitor, idiopathic pulmonary fibrosis, bleomycin, PHMG, EMT

## Abstract

Idiopathic pulmonary fibrosis (IPF) is a fatal lung disease with poor prognosis and progression to lung fibrosis related to genetic factors as well as environmental factors. In fact, it was discovered that in South Korea many people who used humidifier disinfectants containing polyhexamethylene guanidine (PHMG), died of lung fibrosis. Currently two anti-fibrotic drugs, pirfenidone and nintedanib, have been approved by the FDA, but unfortunately, do not cure the disease. Since the histone deacetylase (HDAC) activity is associated with progression to chronic diseases and with fibrotic phenomena in the kidney, heart and lung tissues, we investigated the anti-fibrotic effects of CG-745, an HDAC inhibitor. After lung fibrosis was induced in two animal models by bleomycin and PHMG instillation, the regulation of fibrosis and epithelial mesenchymal transition (EMT)-related markers was assessed. CG-745 exhibited potent prevention of collagen production, inflammatory cell accumulation, and cytokines release in both models. Additionally, N-cadherin and vimentin expression were lowered significantly by the treatment of CG-745. The anti-fibrotic effects of CG-745 proven by the EMT regulation may suggest a potential therapeutic effect of CG-745 on lung fibrosis.

## 1. Introduction 

Idiopathic pulmonary fibrosis (IPF) is the most common and lethal diffuse fibrosis lung disease. It is a chronic, progressive, fibrotic interstitial lung disease of unknown cause that occurs frequently in older adults [1,2]. A favored conceptual model of the IPF pathogenesis is that recurrent, subclinical epithelial injury superimposed on accelerated epithelial aging leads to the aberrant repair of the injured alveolus and deposition of interstitial fibrosis by myofibroblasts. Moreover, the senescence of these alveolar epithelial cells and fibroblasts appears to be a central phenotype that promotes lung fibrosis [3]. 

IPF is generally induced by the usage of antitumor drug, bleomycin, which causes cell damage, oxidative stress, and activates inflammation to develop IPF in animal models [4]. Reportedly, in South Korea, there had been an increasing trend in the incidence of lung fibrosis in recent years which was likely linked to the widespread use of disinfectants for the household humidifier [5,6,7]. According to the Korean Ministry of Environment, women and children were exposed more extensively due to the frequent use and afflicted by the inhalation of chemicals that was known to be PHMG the active ingredient of disinfectants. Various studies have shown the causal association between these lung injuries and the use of disinfectants for the household humidifier and proved that the development of diffuse pulmonary fibrosis was associated with PHMG [7,8].

Although there have been notable advances in the treatment of IPF in recent years, for the treatment of IPF, nintedanib and pirfenidone, are the only ones recommended to use [9]. However, these drugs work to delay the disease progression, not to provide the cure for the disease. Thereby, there is an unmet medical need for an improved drug or for a cure. 

In recent years, several studies have demonstrated that epigenetic modulation regulates expression of genes involved in pathogenesis of the pulmonary fibrosis. It was reported that IPF-fibroblasts exhibit a cancer-like phenotype due to aberrant overexpression of class-I and class-II HDAC enzymes, which appeared to be responsible for their abnormal activation and persistence in IPF, presumably as the result of alterations in the acetylation status of the chromatin and various non-histone proteins [10,11,12].

As an HDAC inhibitor that exhibits potent inhibitory activities against HDAC enzymes, CG-745 has been investigated for a novel molecular targeted anti-cancer therapeutic candidate in pancreatic cancer and myelodysplastic syndrome. Additionally, it was suggested that CG-745 displayed renal protective effects in deoxycorticosterone acetate (DOCA)-salt hypertensive (DSH) rats and anti-fibrotic effects in a renal fibrosis mouse model of unilateral ureteral obstruction (UUO) [13,14]. 

In this study, we investigated whether the inhibitory activities and anti-fibrotic effects of CG-745 show any favorable treatment effect on pulmonary fibrosis in animal models induced by bleomycin and PHMG. 

## 2. Results

### 2.1. Anti-Fibrotic Effect of CG-745 in Bleomycin-Induced Pulmonary Fibrosis in Mice

To evaluate whether CG-745 shows anti-fibrotic effects, mice were induced by 2 mg/kg bleomycin injection through intratracheal instillation. Starting on the next day, CG-745 at a dose of 15 mg/kg, 30 mg/kg, or 60 mg/kg was administered intraperitoneally daily for a total of 14 days (Figure 1A). After sacrificing, morphological changes of the lung by bleomycin injection and CG-745 administration were observed. Alveolar structure and collagen fiber of lung were determined by two different histological methods, hematoxylin-eosin (H&E) staining, and Masson’s trichrome staining, respectively. Bleomycin-induced group showed the expected increase of collagen fiber compared to control. Whereas, CG-745 treated groups reduced the collagen fiber content in a dose-dependent manner (Figure 1B). Thus, bleomycin induced morphological alteration and fibrosis, while administration of CG-745 improved the lung alveolar structure dose-dependently. 

Mice injected with bleomycin showed the expected increase of wet/dry (W/D) weight ratio of lungs compared to the control, indicative of induced inflammation. Treatment with CG-745 reduced the W/D weight ratio of lungs in a dose-dependent manner. More specifically, at a dose of 30 mg/kg of CG-745, the W/D weight of the IPF-induced lungs returned to those of the control (Figure 1C). These results indicate that the inflammation is significantly alleviated by CG-745 administration.

Soluble collagen was assessed by Sircol assay, which measures newly synthesized collagen produced during fibrotic development. As shown in Figure 1D, collagen contents increased significantly in the bleomycin instillation group compared to the control group. CG-745 treatment attenuated the collagen contents in a dose-dependent manner. Treatment with CG-745 at 60 mg/kg lowered the collagen contents back to the levels of the control group. Thus, these results suggest that treatment with CG-745 may work effectively on important factors of IPF; inflammation and collagen content.

### 2.2. The Attenuation of Lung Inflammatory Cells and Fibrosis/EMT-Related Markers by CG745 Administration in Bleomycin-Induced Mice

To determine whether CG-745 have any effect on the bleomycin-induced infiltrated inflammatory cells of the airways and lung parenchyma, the lavage fluids from mice were collected. Once centrifuged, obtained pellets were resuspended in PBS and the total and differential cells in BALF were counted. The number of total inflammatory cells, macrophages, lymphocytes, and neutrophils in BALF of the bleomycin-induced group were significantly elevated compared to those in the control group (Figure 2A). Administration of CG-745 resulted in significantly reduced cell population of BALF compared to those in the bleomycin-induced group, though, not dose-dependently. 

Various cytokine levels in the lavage fluids were analyzed by measuring the concentrations of TNF-α, IL-1β, IL-6, TGF-β1, and IL-13. The expressions for all major cytokines were greatly enhanced in the bleomycin-induced group compared to those of the control group. While the results of the effects for CG-745 treatment groups did not show dose-dependent manner, the reduced cytokine levels (Figure 2B, Appendix A) were clearly seen. Based on these results, we could postulate that CG-745 significantly decreased the inflammation in bleomycin-induced fibrosis, and confirm that CG-745 increased histone acetylation in the bleomycin model as an active HDAC inhibitor. As shown by the Western blot analysis, bleomycin did not induce histone acetylation, but CG-745 administration increased acetylated histone H3 levels in a dose-dependent manner (Figure 3C,D). 

It is known that the injection of bleomycin into mice induces the formation of fibrosis as well as EMT-related markers. Thus, we investigated the effects of CG-745 on fibrosis-related markers, such as α-SMA, collagen I and PAI-1, in addition to EMT markers, such as E-cadherin, N-cadherin and vimentin using real-time PCR and Western blot analysis. As expected, the levels of all fibrotic markers, as well as N-cadherin, were increased in the bleomycin-induced group compared to those in the control group. In contrast, the levels of these fibrotic markers as well as N-cadherin were diminished with the CG-745 administration dose dependently (Figure 3A,C,D). Furthermore, vimentin levels were also effectively reduced by treatment with CG-745 in the bleomycin-induced model (See Appendix A). Vimentin is a well known marker which is associated with EMT-related fibrosis and generally found in various mesenchymal cells.

To further validate the reduction of fibrosis-related markers and EMT markers by CG-745, we performed immunohistochemical analysis with tissues collected from bleomycin-induced and CG-745 treated groups. Indeed, immunohistochemical staining confirmed the changes detected by real-time PCR analysis and Western blot analysis (Figure 3B).

### 2.3. Anti-Fibrotic Effect of CG-745 in PHMG-Induced Pulmonary Fibrosis in Mice

To investigate the effects of CG-745 on PHMG-induced lung fibrosis, 1 mg/kg PHMG was injected into mice through intratracheal instillation. Like the bleomycin-induced model, starting on the next day, CG-745 at a dose of 15 mg/kg, 30 mg/kg, or 60 mg/kg was administered intraperitoneally daily for a total of 14 days (Figure 4A). The PHMG-induced model provoked severe alveolar structure changes and collagen deposition compared to the control group. CG-745 administration showed the improvement on fibrotic lesions and collagen deposition dose-dependently (Figure 4B). 

The assessment of the W/D lung weight ratio between the control and PHMG-induced mice revealed there were higher accumulation of water in the lungs of mice in PHMG-induced like in bleomycin-induced. Once treated those mice with 60 mg/kg of CG-745, the PHMG-induced pulmonary edema was almost reverted to normal status (Figure 4C). 

Additionally, the increased collagen levels were seen among animals in PHMG-induced like in bleomycin-induced and the collagen contents were decreased steeply with the administration of CG-745 (Figure 4D).

To investigate whether PHMG-induced infiltration of inflammatory cells into the airways and parenchyma, the total and differential cells in BALF were measured. The numbers of inflammatory cells, macrophages, and lymphocytes in the BALF of the PHMG-induced group were significantly amplified compared to those in the control group (Figure 5A), while CG-745 administration reduced the cell population in the BALF. The upregulation of cytokines such as TNF-α, IL-1β, IL-6, TGF-β1, and IL-13 was observed in PHMG-induced mice, while the levels of cytokines were clearly reduced in CG-745 administration group, though not in dose-dependent manner (Figure 5B, Appendix A). Based on these results, we conclude that CG-745 treatment clearly attenuates PHMG-induced fibrosis.

### 2.4. Attenuation of Lung Inflammatory Cells and Fibrosis/EMT-Related Markers by CG-745 in PHMG-Induced Mice

The markers such as collagen I, α-SMA, PAI-1, E-cadherin and N-cadherin in animals of PHMG-induced model were examined whether there were any changes of fibrosis and EMT-related markers by CG-745 administration since PHMG instillation provoked a progression to lung fibrosis in mice. PHMG greatly induced the modulation of the above-mentioned markers expression, which was in line with the bleomycin-induced. The expression levels of fibrosis markers were significantly increased by PHMG. The cadherin switch between E-cadherin and N-cadherin was also detected in PHMG-induced model. The expression of fibrosis and EMT markers were reversed by CG-745 administration in a dose-dependent manner (Figure 6). Additionally, the acetylation of histone H3 was increased by CG-745 administration in a dose-dependent manner (Figure 6C,D). Based on the results of these various biomarkers, we can conclude that CG-745 attenuates fibrosis and EMT.

## 3. Discussion

The developmental field of HDAC inhibitors has been mainly focused on anti-cancer therapy because of the strong correlation between HDAC overexpression and cancer initiation/progression. However, recent studies have identified an important relationship between HDAC and non-cancer disease progression, such as neurodegenerative diseases, inflammation, cardiovascular disease, viral infections and so on [15]. It was suggested that epigenetic mechanisms and histone modification strongly influence the phenotype of fibrotic fibroblasts [16]. HDACs were known to induce the secretion of pro-fibrotic cytokines and extracellular matrix formation [17,18]. Additionally, an increased expression of HDACs stimulated fibroblast differentiation into myofibroblasts [19]. Moreover, it was reported that TGF-β induced human alveolar epithelial mesenchymal transition [20], and confirmed that alveolar epithelial cell mesenchymal transition developed in vivo during pulmonary fibrosis [21]. Thus, EMT was regarded as an important step during fibrosis progression.

To investigate the pharmaceutical effects of HDAC inhibitors on fibrosis, Korfei et al. compared the anti-fibrotic effects between pan-HDAC inhibitor, panobinostat, and pirfenidone using in vitro assays with fibroblasts from IPF patients [16]. Pirfenidone is one of the only two approved drugs for the treatment of IPF, as shown slowing down the disease progression rate in clinical trials [22,23,24]. While panobinostat reduced pro-fibrotic phenotypes through cell cycle arrest and apoptosis, pirfenidone reduced pro-fibrotic signaling through STAT-3 inactivation and weak epigenetic alterations in IPF-fibroblasts. 

CG-745 exhibits strong inhibitory activities against Class-I and Class-IIb HDACs. Our compound induced strong cell cycle arrest and apoptosis in pancreatic and renal cells [14,25]. We have not checked yet apoptosis and cell cycle changes in IPF-fibroblast by CG-745. However, based on our results, it can be rationalized that CG-745 as an HDAC inhibitor could induce these anti-fibrotic functions. Moreover, this rationalization was supported by the literatures that reported HDAC inhibitors like SAHA, valproic acid, and trichostatin A were able to regulate tissue fibrosis in various animal studies [12,14,26,27,28]. 

In the present study, we established two different pulmonary fibrosis mouse models induced by bleomycin and PHMG. The bleomycin-induced model has been widely used in the assessment for IPF in vivo, though this model does not fully represent pulmonary fibrosis of humans. The PHMG model was established because PHMG has been known to be linked to the induction of pulmonary inflammation, fibrotic responses, and even deaths in South Korea. Many published literatures have been proved the causal association between aerosol particles from these humidifier disinfectants containing PHMG and the outbreaks of the fibrotic lung diseases and lung injuries, especially among young children and women [5,6,7,8].

Through evaluation of several parameters such as the W/D lung weight ratio, inflammatory cells, cytokine expression, collagen content, EMT marker expression, and histone H3 acetylation levels (Figure 1, Figure 2, Figure 3 and Figure 4), we were able to confirm that bleomycin and PHMG instillation into mice provoked lung fibrosis progression. 

We investigated both mouse models with various concentrations of CG-745 to explore the usage of CG-745 as a new therapeutic option in IPF, and concluded that treatment with CG-745 markedly augmented levels of acetylated histone H3 in a dose-dependent manner in both models through inhibitory activities of HDAC. CG-745 significantly reduced pulmonary edema and the production of collagen I (Figure 1 and Figure 4). Additionally, we observed the reduction in the expression levels of fibrosis and EMT markers in bleomycin and PHMG-induced groups, which was attenuated by CG-745 treatment (Figure 3 and Figure 6).

Increases of inflammatory cells and cytokines involved in pulmonary fibrosis were observed in both mouse models. TGF-β is a well-validated cytokine in fibrosis and differentiates fibroblasts into myofibroblasts by inducing expression of α-SMA [19,20]. Indeed, the increased expression levels of both TGF-β and α-SMA in bleomycin and PHMG-induced models, were significantly decreased by the treatment with CG-745 in both mouse models (Figure 1, Figure 2, Figure 3, Figure 4, Figure 5 and Figure 6). Overall, we conclude that CG-745 revealed strong suppression activities against all upregulated markers, which associated with pulmonary fibrosis. Based on these results, CG-745 could be developed as a promising anti-fibrotic agent for lung fibrosis and IPF.

## 4. Materials and Methods

### 4.1. Materials

Bleomycin was purchased from Carbosynth. (Compton, UK). The PHMG-P solution was obtained from SK Chemicals (Seongnam, South Korea). CG-745 was synthesized by CrystalGenomics (Seongnam, South Korea). Primary antibodies against E-cadherin, H3 and Ac-H3 were purchased from Cell Signaling Technology (Beverly, MA, USA). Antibodies against α-SMA, Collagen I and PAI-1 were purchased from Abcam, Inc. (Cambridge, UK). N-cadherin was purchased from Bioworld Technology (Louis Park, MN, USA). Anti-rabbit IgG-conjugated horseradish peroxidase (HRP) antibody was purchased from Bethyl (Montgomery, TX, USA).

### 4.2. Animals and Treatment

Mice (C57BL/6J) were divided into five groups (n = 15 per group): normal control; bleomycin or PHMG; bleomycin or PHMG + CG-745 15 mg/kg; bleomycin or PHMG + CG-745 30 mg/kg; and bleomycin or PHMG + CG-745 60 mg/kg. Mice were given one intratracheal injection of 2 mg/kg bleomycin or 1 mg/kg PHMG to induce an inflammatory response and fibrosis. After bleomycin or PHMG instillation, CG-745 groups (15 mg/kg, 30 mg/kg, 60 mg/kg, diluted PBS) were treated intraperitoneally for 14 days. These mice were dissected on the 15th day. All procedures were performed in accordance with our institutional animal care and use policies.

### 4.3. Lung wet-to-dry Weight Ratios

The wet/dry (W/D) measurement was used to detect pulmonary edema. At the end of the study, the lungs were collected and weighed immediately to determine the wet weight. Then, after drying the lung tissues in the incubator at 70 °C for 72 h, the dry weight was recorded. 

### 4.4. Bronchoalveolar Lavage Fluid (BALF)

Bronchoalveolar lavage fluid (BALF) was collected by cannulating the trachea and washing the lung with 0.8 mL of sterile saline. BALF was centrifuged at 1200 rpm for 5 min, and the total cell number in BALF was counted using a hemocytometer. Cells were stained with Diff-Quick reagents (Sysmex, Kobe, Japan) and the cell numbers of lymphocytes, neutrophils and alveolar macrophages were counted.

### 4.5. Cytokine Analysis by ELISA

The amounts of IL-1β, IL-6, TNF-α and TGF- β1 in the BALF were analyzed by ELISA mouse IL-6 (BD Bioscience, San Jose, CA, USA), IL-1β (BD Bioscience, San Jose, CA, USA), TGF- β1 (eBioscience, San Diego, CA, USA) and TNF-α (eBioscience, San Diego, CA, USA) kits according to the manufacturer’s instructions.

### 4.6. Quantification of Soluble Collagen

Total lung collagen in the BALF was quantified using the Sircol collagen assay (Biocolor Ltd., Newtownabbey, UK), according to the manufacturer’s instructions.

### 4.7. Histological Analysis

At the end of the study, lungs of the mice were harvested and fixed in 100 mL of 4% paraformaldehyde for 24 h. The fixed lungs were then sliced midsagittally and embedded in paraffin. Sections (4 μm) were stained with Masson’s Trichrome staining (Abcam, Cambridge, UK) or Hematoxyline and Eosin (H&E) staining (Hematoxyline; Invitrogen, Carlsbad, CA/Eosin; Sigma-Aldrich, Louis, MO, USA) for morphological analysis and evaluation of collagen deposition.

### 4.8. Immunohistochemistry

Tissue sections were deparaffinized and rehydrated using a graded ethanol series. After antigen retrieval, eliminating endogenous peroxidase and pre-incubating with 5% normal goat serum to block background staining, sections were incubated with rabbit anti-α-SMA antibody (1:200) or rabbit anti-Collagen I antibody (1:100) or rabbit anti-N-cadherin antibody (1:100) or rabbit anti-E-cadherin antibody (1:50) at 4 °C overnight. The color reaction was then performed with HRP-linked polymer detection system and immunovisualization was carried out with 3,3′- diaminobenzidine as substrate (Sigma, St Louis, MO, USA).

### 4.9. Real-Time Polymerase Chain Reaction (PCR) Analysis

Total RNA from mouse lung tissues were extracted using an easy-BLUE Total RNA Extraction Kit (Intron, Seongnam, Republic of KOREA) according to the protocol provided by the manufacturer. A 1μg sample of RNA was reverse transcribed to complementary DNA using SuperScript II First-Strand Synthesis SuperMix for qRT-PCR (Invitrogen, Carlsbad, CA, USA), following the company’s instructions. Real-time PCR was performed on complementary DNA (cDNA) samples using the SYBR Green system (Bioneer, Daejeon, Republic of KOREA). Used primers were α-SMA, sense 5′-CTGACAGAGGCACCACTGAA-3′ and anti-sense 5′-CATCTCCAGAGTCCAGCACA-3′, Collagen I, sense 5′-ACGGCTGCACGAGTCACAC-3′ and anti-sense 5′-GGCAGGCGGGAGGTCTT-3′, E-cadherin, sense 5′-GGTTTTCTACAGCATCACCG-3′ and anti-sense 5′-GCTTCCCCATTTGATGACAC-3′, N-cadherin, sense 5′-TGAAACGGCGGGATAAAGAG-3′ and anti-sense 5′-GGCTCCACAGTATCTGGTTG-3′, GAPDH, sense 5′-AGGTCGGTGTGAACGGATTTG-3′ and anti-sense 5′-GGCCTCACCCCATTTGATGT-3′. The following general real-time PCR protocol was employed: denaturation for 10 min at 95 °C, followed by 40 cycles of denaturation at 95 °C for 10 s, annealing at 60 °C for 30 s and extension at 72 °C for 1 sec with a single fluorescence acquisition step at the end of extension, a melting step by slow heating from 60 °C to 99 °C with a rate of 0.1 °C/s and continuous fluorescence measurement, and a final cooling step to 40 °C [29]. Crossing point values were acquired by using the second derivative maximum method of the LightCycler software 3.3 (Roche, Burlington, NC, USA). Real-time PCR efficiencies were acquired by amplification of a standardized dilution series, and slopes were determined using LightCycler software (Roche, Burlington, NC, USA).

### 4.10. Western Blotting

Lung tissues were harvested and lysed in RIPA Lysis buffer (ATTO, Tokyo, Japan) for 20 min on ice. After centrifugation at 12,000× *g* for 30 min at 4 °C, and the supernatant was mixed with one-fifth volume of 5× sodium dodecyl sulfate (SDS) sample buffer, boiled for 5 min, and then separated by a 10% SDS-polyacrylamide electrophoresis gel. After electrophoresis, proteins were transferred to a polyvinylidenedifluoride (PVDF) membranes, which were blocked in TBS-T (25 mM Tris [pH, 7.6], 138 mMNaCl, and 0.05% Tween-20) containing 5% skim milk. Membranes were incubated with primary antibodies (at 1:1000–1:5000). After a series of washes, the membranes were further incubated with secondary antibody conjugated with HRP (at 1:2000–1:10,000). The immunoreactive signal was detected using an ECL detection system (Amersham, Buckinghamshire, England, UK). 

### 4.11. Statistical Analysis 

Each experiment was performed at least three times, and all values were expressed as the mean ± SD of triplicate samples. The Student’s *t*-test was used to determine statistical significance. Values of *p* < 0.05 were considered statistically significant.

## Figures and Tables

**Figure 1 molecules-24-02792-f001:**
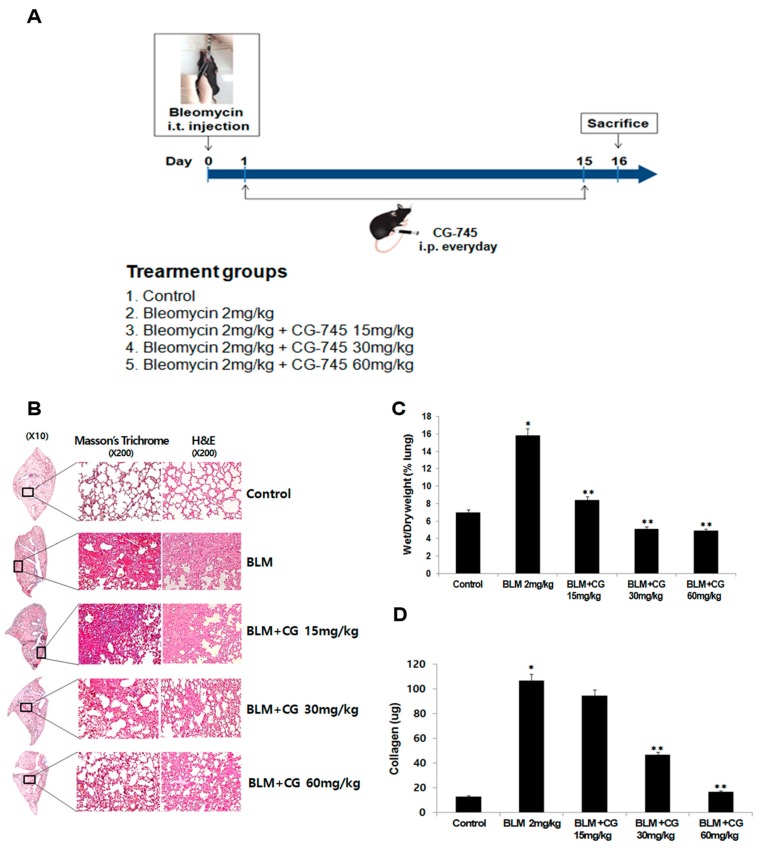
CG-745 ameliorates pulmonary fibrosis induced by bleomycin in mice. (**A**) Mice were instilled with bleomycin (2 mg/kg) intratracheally on day zero, and CG-745 (15 mg/kg, 30 mg/kg or 60 mg/kg) was intraperitoneally administered daily for 14 days. (**B**) Representative histological sections from lung tissues stained with hematoxylin and eosin (H&E) or Masson’s trichrome staining were shown from the control, bleomycin, and bleomycin + CG-745 (15 mg/kg, 30 mg/kg, or 60 mg/kg) group (magnification, ×200). (**C**) The wet/dry measurement was used to detect pulmonary edema. (**D**) Collagen content in the lung tissues was measured by a Sircol assay. Data are presented as mean ± SD, n = 5. * *p* < 0.01 versus control group. ** *p* < 0.05 versus bleomycin group.

**Figure 2 molecules-24-02792-f002:**
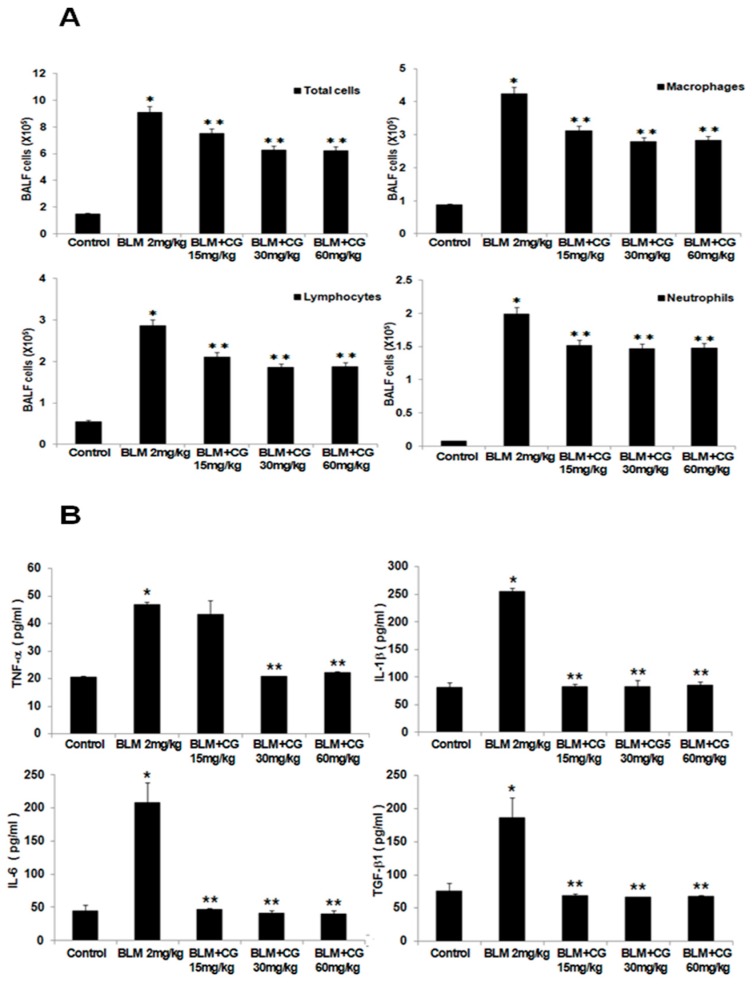
CG-745 ameliorates pulmonary inflammation induced by bleomycin in mice. (**A**) The number of total cells, macrophages, neutrophils, and lymphocytes in the bronchoalveolar lavage fluid (BALF) was counted. (**B**) The levels of TNF-α, IL-1β, IL-6 and TGF-β1 in the BALF were analyzed by ELISA. Data are presented as mean ± SD, n = 5. * *p* < 0.01 versus control group. ** *p* < 0.05 versus bleomycin group.

**Figure 3 molecules-24-02792-f003:**
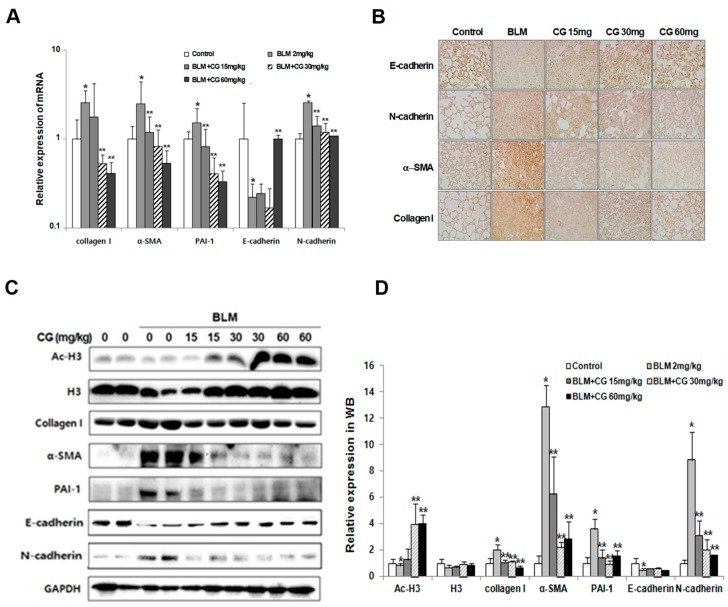
Effects of CG-745 on bleomycin-induced epithelial-mesenchymal transition and fibrosis of mice. (**A**) The mRNA (messenger RNA) levels of fibrotic marker (collagen I, α-SMA, PAI-1) and EMT (Epithelial-Mesenchymal Transition) marker (E-cadherin, N-cadherin) expressions in the lung tissues were analyzed by real-time qPCR. Gene expression was presented as fold changes and normalized to GAPDH (Glyceraldehyde-3-Phosphate Dehydrogenase). (**B**) Representative immunohistochemical staining of α-SMA, collagen I, E-cadherin and N-cadherin in mice (magnification, ×200). (**C**) The protein levels of fibrotic marker (collagen I, α-SMA, PAI-1) and EMT marker (E-cadherin, N-cadherin), H3 (histone H3) and Ac-H3 (acetylated histone H3) in the lung tissues were analyzed by western blotting. Immunoblots are representative of at least three independent experiments. (**D**) The bar graphs for an average (n = 15/group) shows for providing a quantitative data. Data are presented as mean ± SD, n = 5. * *p* < 0.01 versus control group. ** *p* < 0.05 versus bleomycin group.

**Figure 4 molecules-24-02792-f004:**
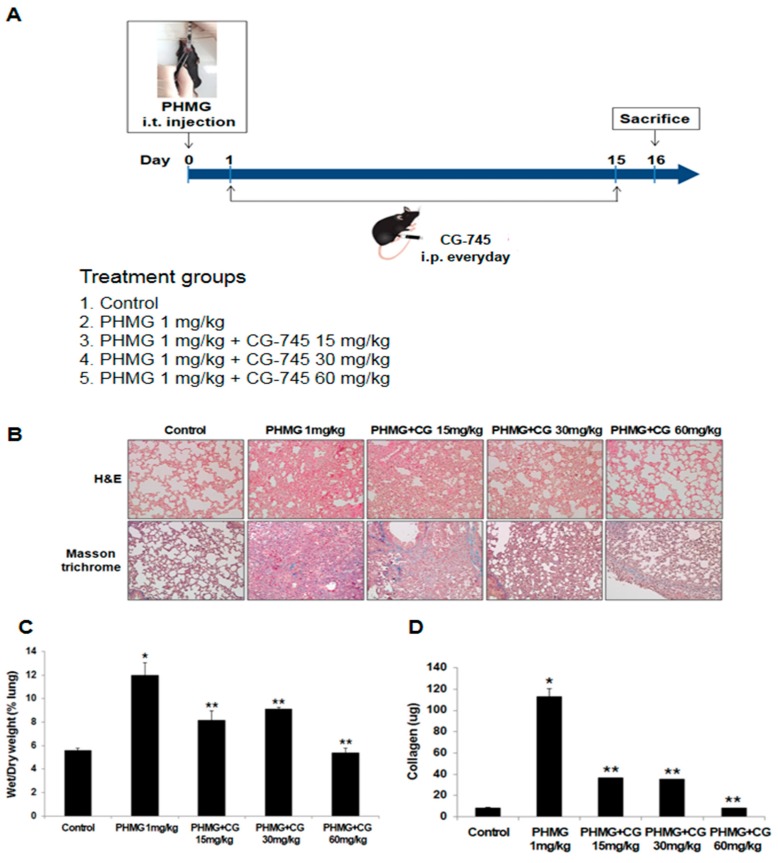
CG-745 ameliorates pulmonary fibrosis induced by PHMG in mice. (**A**) Mice were instilled with polyhexamethylene guanidine (PHMG) (1 mg/kg) intratracheally on day zero only, and then CG-745 (15 mg/kg, 30 mg/kg or 60 mg/kg) was intraperitoneally administered daily for 14 days. (**B**) Representative histological sections from lung tissues stained with hematoxylin and eosin (H&E) or Masson’s trichrome staining were shown from control, PHMG, and PHMG + CG-745 (15 mg/kg, 30 mg/kg, or 60 mg/kg) group (magnification, ×200). (**C**) The wet/dry measurement was used to detect pulmonary edema. (**D**) Collagen content in the lung tissues was measured by a Sircol assay. Data are presented as mean ± SD, n = 5. * *p* < 0.01 versus control group. ** *p* < 0.05 versus PHMG group.

**Figure 5 molecules-24-02792-f005:**
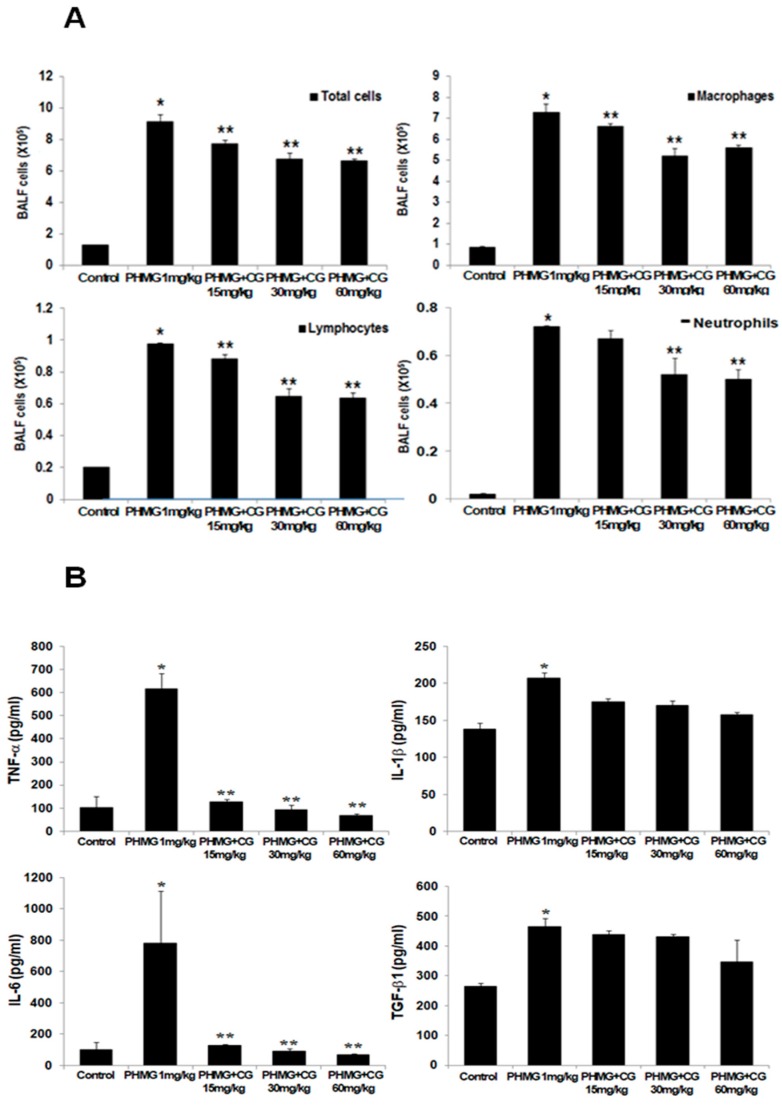
CG-745 ameliorates pulmonary inflammation induced by PHMG in mice. (**A**) The number of total cells, macrophages, neutrophils, and lymphocytes in the bronchoalveolar lavage fluid (BALF) was counted. (**B**) The levels of TNF-α, IL-1β, IL-6 and TGF-β1 in the BALF were analyzed by ELISA. Data are presented as mean ± SD, n = 5. * *p* < 0.01 versus control group. ** *p* < 0.05 versus PHMG group.

**Figure 6 molecules-24-02792-f006:**
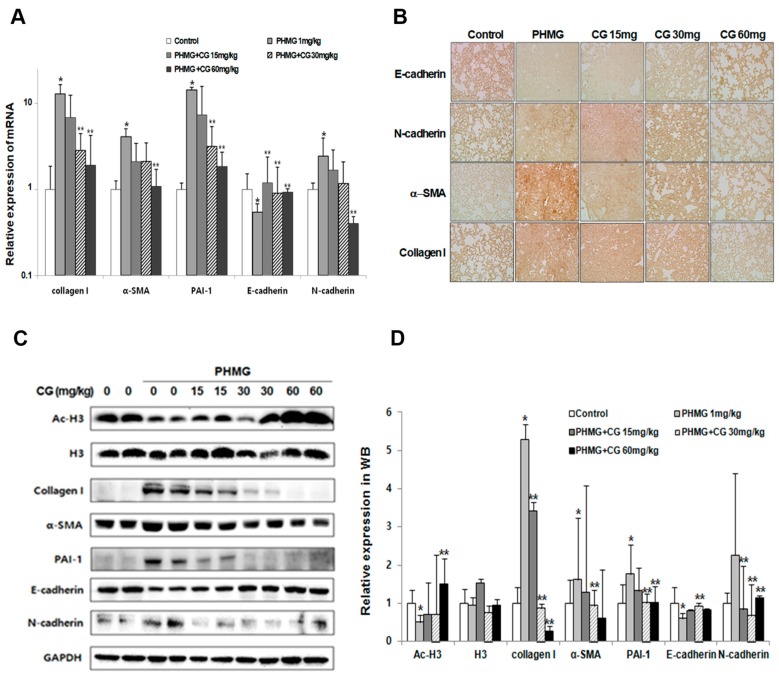
Effects of CG-745 on PHMG-induced epithelial-mesenchymal transition and fibrosis of mice. (**A**) The mRNA levels of fibrotic marker (collagen I, α-SMA, PAI-1) and EMT marker (E-cadherin, N-cadherin) expressions in the lung tissues were analyzed by real-time qPCR. Gene expression was presented as fold changes and normalized to GAPDH. (**B**) Representative immunohistochemical staining of collagen I, α-SMA, E-cadherin and N-cadherin in mice (magnification, ×200). (**C**) The protein levels of fibrotic marker (collagen I, α-SMA, PAI-1), EMT marker (E-cadherin, N-cadherin), H3 (histone H3) and Ac-H3 (acetylated histone H3) in the lung tissues were analyzed by western blotting. Immunoblots are representative of at least three independent experiments. (**D**) The bar graphs for an average (n = 15/group) shows for providing a quantitative data. Data are presented as mean ± SD, n = 5. * *p* < 0.01 versus control group. ** *p* < 0.05 versus PHMG group.

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
