# Peer review of "The Anti-Fibrotic Effects of CG-745, an HDAC Inhibitor, in Bleomycin and PHMG-Induced Mouse Models"

_molecules, 2019, doi:10.3390/molecules24152792_

Round 1

Reviewer 1 Report

The manuscript entitled "The anti-fibrotic effects of CG-745, an HDAC  inhibitor, in bleomycin and PHMG-induced mouse  models" provides quality results.

I also described some parts that could be improved where the sentences and paragraphs are even more difficult to follow.

1. The chapter 2.3 needs to be reformulate...the investigation method is not clear formulated. Please give more details. Did you used the PHMG intratracheal instillation just on day 0 or for all 14 days? The figure's no. 4 legend is also confusing, please reformulate.

2. When you use abbreviations, for the first time in your manuscript, you have to give details about that.

3. Row no 96, 162 please capitalize the word "masson's".

4. Discussion chapter- row no 230,231- can you explain why did you used this affirmation...in your manuscript you have not tested any effect on the renal cells or the pancreatic cells... please give details about this aspects

Thank You!  

Reviewer 2 Report

In this paper, the authors investigated the potential effects of an HDAC inhibitor, 

CG-745 in diopathic pulmonary fibrosis by using animal model. The experiment was well performed and the conclusion is solid.  I only have a few minor concerns.

1. for the western blot shown in fig. 3 and fig. 6, how many times did they perform? The number underneath each blot is quantified from one blot or an average? I hope they could provide a quantitative data.

2. for real time PCR, I feel they need to provide more clear and accurate information.

why final cooling step to 40°C? how do they normalize the data?

Reviewer 3 Report

This is an interesting paper, the authors look at CG-745 as an histone deactylase inhibitor in bleomycin induced fibrosis and another model.   This is very interesting and appears to show reduction in fibrosis associated with a reduction in inflammatory cells and inflammatory cytokines. The authors looked at IL-1beta, IL-6, TNF-alpha and Transforming growth factor beta.  The authors must also look in the BAL for the pro-fibrotic Interleukin-13 (IL-13).  IL-13 is a very potent pro-fibrotic molecule.

Also in terms of mechanism the authors must also look at Gremlin-1 which is also a very potent pro-fibrotic molecule. They must look at the expression of Gremlin-1 with bleomycin and bleomycin + CG-745 treatment.

Round 2

Reviewer 3 Report

This is now acceptable.